# Performance Comparison of Lattice-Matched AlInN/GaN/AlGaN/GaN Double-Channel Metal–Oxide–Semiconductor High-Electron Mobility Transistors with Planar Channel and Multiple-Mesa-Fin-Channel Array

**DOI:** 10.3390/ma15010042

**Published:** 2021-12-22

**Authors:** Hsin-Ying Lee, Ying-Hao Ju, Jen-Inn Chyi, Ching-Ting Lee

**Affiliations:** 1Department of Photonics, National Cheng Kung University, Tainan 701, Taiwan; hylee@ee.ncku.edu.tw; 2Department of Electrical Engineering, National Central University, Zhongli 32001, Taiwan; s17howard1001@gmail.com (Y.-H.J.); chyi@ee.ncu.edu.tw (J.-I.C.); 3Department of Electrical Engineering, Yuan Ze University, Taoyuan 320, Taiwan

**Keywords:** double-channel epitaxial structure, double-hump transconductance, Ga_2_O_3_ gate oxide layer, metal–oxide–semiconductor high-electron mobility transistors

## Abstract

In this work, Al_0.83_In_0.17_N/GaN/Al_0.18_Ga_0.82_N/GaN epitaxial layers used for the fabrication of double-channel metal–oxide–semiconductor high-electron mobility transistors (MOSHEMTs) were grown on silicon substrates using a metalorganic chemical vapor deposition system (MOCVD). A sheet electron density of 1.11 × 10^13^ cm^−2^ and an electron mobility of 1770 cm^2^/V-s were obtained. Using a vapor cooling condensation system to deposit high insulating 30-nm-thick Ga_2_O_3_ film as a gate oxide layer, double-hump transconductance behaviors with associated double-hump maximum extrinsic transconductances (g_mmax_) of 89.8 and 100.1 mS/mm were obtained in the double-channel planar MOSHEMTs. However, the double-channel devices with multiple-mesa-fin-channel array with a g_mmax_ of 148.9 mS/mm exhibited single-hump transconductance behaviors owing to the better gate control capability. Moreover, the extrinsic unit gain cutoff frequency and maximum oscillation frequency of the devices with planar channel and multiple-mesa-fin-channel array were 5.7 GHz and 10.5 GHz, and 6.5 GHz and 12.6 GHz, respectively. Hooge’s coefficients of 7.50 × 10^−5^ and 6.25 × 10^−6^ were obtained for the devices with planar channel and multiple-mesa-fin-channel array operating at a frequency of 10 Hz, drain–source voltage of 1 V, and gate–source voltage of 5 V, respectively.

## 1. Introduction

In recent decades, silicon (Si)-based electronic devices have become dominant power devices used in various systems. To enhance their capability of operating at a higher voltage, higher current, higher temperature, higher frequency, and with better energy efficiency, gallium nitride (GaN)-based electronic devices have played the mainstream role of power semiconductor devices, which have recently benefited from their superior electrical and physical properties [1,2,3]. Despite the success of depletion-mode and enhancement-mode GaN-based single-channel metal–oxide–semiconductor high-electron mobility transistors (MOSHEMTs), active development of high-performance compelling devices is still needed. In general, the enhancement of both electron mobility (μ_n_) and sheet electron density (n_ch_) has been widely explored with the aim of improving the performances of GaN-based MOSHEMTs. However, it is technically difficult to grow an AlGaN layer with high Al content to achieve high μ_n_ and n_ch_ simultaneously. Recently, to simultaneously obtain both the enhanced μ_n_ and n_ch_, multiple-channel structures were explored [4,5,6,7]. Furthermore, superior performances of higher current drive, low resistance, low-frequency noise, and improved linearity were demonstrated in multiple-channel MOSHEMTs [8,9,10]. Nevertheless, since lattice-matched heterostructured structures could reduce the failure caused by the inverse piezoelectric effect [11], lattice-matched barrier layers in multiple-channel structures remain a promising candidate for enhancing performance [12,13,14]. In this work, double-channel epitaxial layers of lattice-matched Al_0.83_In_0.17_N/GaN/Al_0.18_Ga_0.82_N/GaN were grown on Si substrates using a metalorganic chemical deposition (MOCVD, AIXTRON Group, Herzogenrath, Germany) system. Although several gate oxide layers were used in GaN-based MOSHEMTs [15,16,17,18,19,20], gallium oxide (Ga_2_O_3_)-based materials have become promising gate oxide layers due to their superior properties of high breakdown voltage, high radiation resistance, high thermal and chemical stability, high Baliga’s figure-of-merit, and better interface properties between Ga_2_O_3_ film and GaN-based semiconductors [21,22,23]. Furthermore, because high-quality and high-insulating amorphous Ga_2_O_3_ films could be deposited using a vapor cooling condensation system [24,25] and were successfully used in GaN-based MOSHEMTs previously [25,26], the system was used to deposit a 30-nm-thick Ga_2_O_3_ film as a gate oxide layer in this work. In addition, to improve interface properties, surface preparation was employed previously [27]. In this study, before depositing the Ga_2_O_3_ gate oxide layer, an (NH_4_)_2_S_x_ chemical solution was used to treat the sample surface to completely remove the undesired native oxide residing on the surface of the GaN-based semiconductors [28]. Recently, lattice-matched double-channel AlInN/GaN/AlGaN/GaN MOSHEMTs with multiple-mesa-fin-channel array were reported [29]. Although the compared performances of signal-channel MOSHEMTs with planar channel and multiple-mesa-fin-channel array were reported previously [30,31], a comparison of the performances of the multiple-channel MOSHEMTs with planar channel and multiple-mesa-fin-channel array has not yet been carried out. In this work, to compare the performances of the planar channel structure with the multiple-mesa-fin-channel array of double-channel MOSHEMTs, lattice-matched double-channel Al_0.83_In_0.17_N/GaN/Al_0.18_Ga_0.82_N/GaN MOSHEMTs with planar channel were fabricated and studied.

## 2. Epitaxial Growth and Results

A MOCVD system was utilized to grow epitaxial layers of Al_0.83_In_0.17_N/GaN/Al_0.18_Ga_0.82_N/GaN double-channel structure on Si substrate. The epitaxial structure included an AlN nucleation layer (250 nm), a graded AlGaN buffer layer (1.1 μm), a trimethylgallium (TMGa)-grown undoped GaN buffer layer (1.9 μm), a triethylgallium (TEGa)-grown GaN channel 1 layer (100 nm), an AlN spacer layer (1 nm), an Al_0.18_Ga_0.82_N barrier layer 1 (25 nm), a TEGa-grown GaN channel 2 layer (10 nm), an AlN spacer layer (1 nm), an Al_0.83_In_0.17_N barrier layer 2 (8 nm), and a GaN cap layer (2 nm). In the epitaxial growth, trimethylgallium (TMAl) and trimethylindium (TMIn) were used as the precursors for Al and In sources to grow the Al_0.18_Ga_0.82_N barrier layer and theAl_0.83_In_0.17_N barrier layer, respectively. In addition, ammonia (NH_3_) and hydrogen (H_2_) were respectively used as the nitrogen source and carrier gas for growing the 1.1-μm-thick graded AlGaN buffer layer and 1.9-μm-thick GaN buffer layer. The H_2_ carrier gas was replaced by nitrogen gas just before and at the end of the growth of the GaN buffer layer without growth interruption. Furthermore, by using a TEGa precursor to grow the GaN channel, the residual carbon concentration could be reduced to improve electron mobility due to the lower channel trapping [32].

Regarding the simulation of a one-dimensional (1D) Schrödinger–Poisson solver, double two-dimensional electron gas (2-DEG) channels were formed at the polarized Al_0.18_Ga_0.82_N/GaN interface and the band-discontinued lattice-matched Al_0.83_In_0.17_N/GaN interface [29]. Using a Hall measurement (Ecopia Crop., Anyang, Korea) at room temperature, the equivalent electron mobility of 1770 cm^2^/V-s and the equivalent sheet electron density of 1.11 × 10^13^ cm^−2^ were measured. To compare the performances of double-channel MOSHEMTs with planar channel and multiple-mesa-fin-channel array, the same epitaxial layers were used in this work.

## 3. Results and Discussion

Figure 1a,b present the 3D schematic configuration of the double-channel Al_0.83_In_0.17_N/GaN/Al_0.18_Ga_0.82_N/GaN MOSHEMTs with planar channel and multiple-mesa-fin-channel array, respectively. Under the protection of a patterned 500-nm-thick Ni metal mask, the mesa isolation region (310 μm × 320 μm) of MOSHEMTs was etched down to Si substrate using a reactive-ion etching system with a BCl_3_ etchant. After removing the Ni mask using an HCl chemical solution, the sample surface was treated by an (NH_4_)_2_S_x_ chemical solution at 60 °C for 30 min to completely remove undesired native oxide. Using a standard photolithography system to open windows of the source electrode and drain electrode, laminated Ti/Al/Pt/Au (25/100/50/400 nm) metals were sequentially deposited and were then lifted off. By annealing the sample in a nitrogen environmental rapid-thermal annealing system at 850 °C for 1 min, an ohmic-contacted source electrode and drain electrode with a specific contact resistance of about 7.5 × 10^−^^6^ Ω-cm^2^ were obtained. The separation between the source electrode and the drain electrode was 10 μm. After depositing a ZnO film (300 nm), the gate windows (length = 1 μm and width = 50 μm) placed at the central region between the source electrode and drain electrode were etched using a diluted HCl chemical solution under a photoresist mask. When the sample was repeatedly treated with a (NH_4_)_2_S_x_ chemical solution, a 30-nm-thick Ga_2_O_3_ film was deposited at about 80 K as the gate oxide layer using a vapor cooling condensation system. The deposition process and electrical properties of the amorphous Ga_2_O_3_ films deposited using this system were reported previously [25,33]. Prior to removing the ZnO mask, laminated Ni/Au (20/300 nm) gate metals were sequentially deposited by an electron beam evaporator. The double-channel planar Al_0.83_In_0.17_N/GaN/Al_0.18_Ga_0.82_N/GaN MOSHEMTs have a channel length of 10 μm and a channel width of 50 μm. In the same batch process, two samples of the double-channel MOSHEMTs with planar channel and multiple-mesa-fin-channel array were fabricated. Every sample included more than 200 devices. The uniformity and performance reproducibility of the MOSHEMTs were good.

In addition to the 500-nm-wide strip channel array patterned using an He-Cd laser interference photolithography system, similar fabrication processes of planar devices were utilized for fabricating devices with multiple-mesa-fin-channel array. However, within the same channel width of 50 μm, the total real channel width in the multiple-mesa-fin-channel array was 25.2 μm [29].

Using the measurement of an Agilent 4156C semiconductor parameter analyzer, Figure 2a,b respectively illustrate the typical drain–source current (I_DS_)−drain–source voltage (V_DS_) characteristics of the devices with planar channel and multiple-mesa-fin-channel array operating at various gate–source voltages (V_GS_). By normalizing the channel width of 50 μm, the normalized saturation drain–source current (I_DSS_) of the planar devices operating at V_GS_ = 5 V and V_DS_ = 10 V was 520.0 mA/mm, in which the real drain–source current was 26.0 mA. For the devices with multiple-mesa-fin-channel array, the I_DSS_ was 842.7 mA/mm, in which the real drain–source current was 21.2 mA. It was worth noting that the real drain–source current in the devices with multiple-mesa-fin-channel array was less than that of the planar devices. When the on-resistance (R_on_) of the devices was defined as the inverse slope of the I_DS_−V_DS_ characteristics at V_DS_ = 0 V and V_GS_ = 5 V, the R_on_ = 10.2 Ω-mm of the planar devices was calculated and compared with the R_on_ = 6.1 Ω-mm of the devices with multiple-mesa-fin-channel array [29].

Figure 3 presents the I_DS_−V_GS_ characteristics and the extrinsic transconductance (g_m_)−V_GS_ characteristics of the devices with planar channel and multiple-mesa-fin-channel array operating at V_DS_ = 10 V. In the g_m_−V_GS_ characteristics shown in Figure 3, a double-hump transconductance performance was clearly observed in the planar devices; it corresponded to the effective gate modulation of the upper channel 2 and the lower channel 1, respectively. The maximum extrinsic transconductance (g_mmax_) of the double-hump behavior was 89.8 and 100.1 mS/mm, respectively. It was worth noting that the double-hump transconductance performance was not found in the devices with a multiple-mesa-fin-channel structure due to the better gate control capability caused by the modulation of the side-wall electrical field in the fin-channel [31]. In addition, the g_mmax_ of 148.9 mS/mm was obtained in the single-hump transconductance. Compared with the performances of the devices with planar channel, the higher normalized saturation current, higher maximum extrinsic transconductance, and lower on-resistance in the devices with multiple-mesa-fin-channel array were contributed by the lower real drain–source current and the better heat dissipation driven by lateral heat flow within the fin-channel array [31]. If the threshold voltage (V_th_) of the devices was defined as the V_GS_ corresponding to the I_DS_ = 1 μA/mm, the V_th_ of the planar devices was −3.8 V. Compared with a V_th_ of −3.2 V in the devices with a multiple-fin-channel array structure, the V_th_ was pushed toward a more positive value due to the early pinched-off effect [30,34].

An Agilent 8510C network analyzer (Agilent, CA, USA) was used to measure small-signal high-frequency performances of the MOSHEMTs. Figure 4 illustrates frequency-dependent short-circuit current gain and frequency-dependent maximum available power gain of the devices with planar channel and multiple-mesa-fin-channel array. The extrinsic unit gain cutoff frequency (f_T_) of 5.7 GHz and the maximum oscillation frequency (f_max_) of 10.5 GHz of the planar devices were obtained. For the MOSHEMTs with multiple-mesa-fin-channel array, the f_T_ and f_max_ were 6.5 GHz and 12.6 GHz, respectively [29].

In general, the low-frequency noise behaviors could be used to evaluate electron trapping and electron detrapping effects induced by traps, defects, and interface states in electronic devices [35]. The low-frequency noise characteristics of both the devices were measured using an Agilent 4156C low-noise bias supply, an HP 35670A dynamic signal analyzer, and a BTA 9812B noise analyzer. Figure 5 presents the normalized low-frequency noise power density $\left(S_{\mathrm{IDS}}\left(f\right)/I_{\mathrm{DS}}{}^{2}\right)$−frequency (f) characteristics of the devices with planar channel and multiple-mesa-fin-channel array operating at V_DS_ = 1 V. As shown in Figure 5, the normalized noise power density decreased with an increase in V_GS_ voltage. The $S_{\mathrm{IDS}}\left(f\right)/I_{\mathrm{DS}}{}^{2}$ was about 5.4 × 10^−13^ Hz^−1^ and 8.7 × 10^−14^ Hz^−1^ for the devices with planar channel and multiple-mesa-fin-channel array operating at V_GS_ = 5 V and V_DS_ = 1 V at f = 10 Hz. Under the same operating conditions, the higher normalized noise power density in the planar devices was attributed to the larger channel area. Generally, Hooge’s coefficient α provided a useful figure-of-merit for evaluating electronic devices. The α value could be calculated as follows [36]:(1)$$\alpha =\left(S_{\mathrm{IDS}}\left(f\right)/I_{\mathrm{DS}}{}^{2}\right)\cdot f\cdot \left(L_{G}W_{G}n_{\mathrm{ch}}\left(V_{\mathrm{GS}}-V_{\mathrm{th}}\right)/\left|V_{\mathrm{th}}\right|\right)$$
where L_G_ = 1 μm is gate length and n_ch_ = 1.11 × 10^13^ cm^−2^ is channel electron density. In the planar devices, the gate width W_G_ is 50 μm and the threshold voltage V_th_ is −3.8 V, while W_G_ is 25.2 μm and V_th_ is −3.2 V in the devices with multiple-mesa-fin-channel array. The calculated α value of 7.50 × 10^−5^ and 6.25 × 10^−6^ for the MOSHEMTs with planar channel and multiple-mesa-fin-channel array operating at f = 10 Hz, V_GS_ = 5 V, and V_DS_ = 1 V was obtained. In addition to the result of low Hooge’s coefficient, the normalized noise power density was well fixed at 1/f. This behavior could imply that the flicker noise was the dominant noise in the devices.

## 4. Conclusions

Recently, planar channel structures and multiple-mesa-fin-channel array structures have been widely used for the fabrication of MOSHEMTs. In this study, to compare the performance of planar channel and multiple-mesa-fin-channel array used in double-channel MOSHEMTs, double-channel Al_0.83_In_0.17_N/GaN/Al_0.18_Ga_0.82_N/GaN epitaxial layers were grown on Si substrates using an MOCVD system. The performance comparisons of the double-channel MOSHEMTs with planar channel and multiple-mesa-fin-channel array are listed in Table 1. Compared with the performance of the double-channel MOSHEMTs with multiple-mesa-fin-channel array, double-hump transconductance behavior was observed in the double-channel planar MOSHEMTs owing to the effective gate modulation of the upper channel 2 and the lower channel 1 by the only top-side electrical field. However, in addition to the modulation of the top-side electrical field, the side-wall electric field in the multiple-mesa-fin-channel could perform at a better gate control capability. Consequently, the transconductance collapse was paved in the multiple-mesa-fin-channel. It was expected that the pavement efficiency could be enhanced by using narrower fin-channel. The extrinsic maximum double-hump transconductances of the double-channel planar MOSHEMTs were 89.8 and 100.1 mS/mm, respectively. In addition, the extrinsic maximum transconductance of the double-channel MOSHEMTs with multiple-mesa-fin-channel array was 148.9 mS/mm. Furthermore, compared with the performances of MOSHEMTs with a planar channel structure, better heat dissipation driven by the lateral heat flow in the parallel fin-channels could improve the direct-current performances and high-frequency performances [37]. The extrinsic unit gain cutoff frequency and maximum oscillation frequency of the MOSHEMTs with planar channel and multiple-mesa-fin-channel array were 5.7 and 10.5 GHz, and 6.5 and 12.6 GHz, respectively. Owing to the screening effect of the trapping probability, better noise performances were obtained in the MOSHEMTs with multiple-mesa-fin-channel array [31,38]. Hooge’s coefficient of the MOSHEMTs with planar channel and multiple-mesa-fin-channel array was 7.50 × 10^−5^ and 6.25 × 10^−6^, respectively. Although the multiple-mesa-fin-channel array exhibited superior advantages to the MOSHEMTs, its real drain–source current was less than that in the planar devices with the same channel width. Evaluating the power performances of devices with multiple-mesa-fin-channel array remains a challenging task.

Recently, planar channel structures and multiple-mesa-fin-channel array structures have been widely used for the fabrication of MOSHEMTs. In this study, to compare the performance of planar channel and multiple-mesa-fin-channel array used in double-channel MOSHEMTs, double-channel Al_0.83_In_0.17_N/GaN/Al_0.18_Ga_0.82_N/GaN epitaxial layers were grown on Si substrates using an MOCVD system. The performance comparisons of the double-channel MOSHEMTs with planar channel and multiple-mesa-fin-channel array are listed in Table 1. Compared with the performance of the double-channel MOSHEMTs with multiple-mesa-fin-channel array, double-hump transconductance behavior was observed in the double-channel planar MOSHEMTs owing to the effective gate modulation of the upper channel 2 and the lower channel 1 by the only top-side electrical field. However, in addition to the modulation of the top-side electrical field, the side-wall electric field in the multiple-mesa-fin-channel could perform at a better gate control capability. Consequently, the transconductance collapse was paved in the multiple-mesa-fin-channel. It was expected that the pavement efficiency could be enhanced by using narrower fin-channel. The extrinsic maximum double-hump transconductances of the double-channel planar MOSHEMTs were 89.8 and 100.1 mS/mm, respectively. In addition, the extrinsic maximum transconductance of the double-channel MOSHEMTs with multiple-mesa-fin-channel array was 148.9 mS/mm. Furthermore, compared with the performances of MOSHEMTs with a planar channel structure, better heat dissipation driven by the lateral heat flow in the parallel fin-channels could improve the direct-current performances and high-frequency performances [37]. The extrinsic unit gain cutoff frequency and maximum oscillation frequency of the MOSHEMTs with planar channel and multiple-mesa-fin-channel array were 5.7 and 10.5 GHz, and 6.5 and 12.6 GHz, respectively. Owing to the screening effect of the trapping probability, better noise performances were obtained in the MOSHEMTs with multiple-mesa-fin-channel array [31,38]. Hooge’s coefficient of the MOSHEMTs with planar channel and multiple-mesa-fin-channel array was 7.50 × 10^−5^ and 6.25 × 10^−6^, respectively. Although the multiple-mesa-fin-channel array exhibited superior advantages to the MOSHEMTs, its real drain–source current was less than that in the planar devices with the same channel width. Evaluating the power performances of devices with multiple-mesa-fin-channel array remains a challenging task.

## Figures and Tables

**Figure 1 materials-15-00042-f001:**
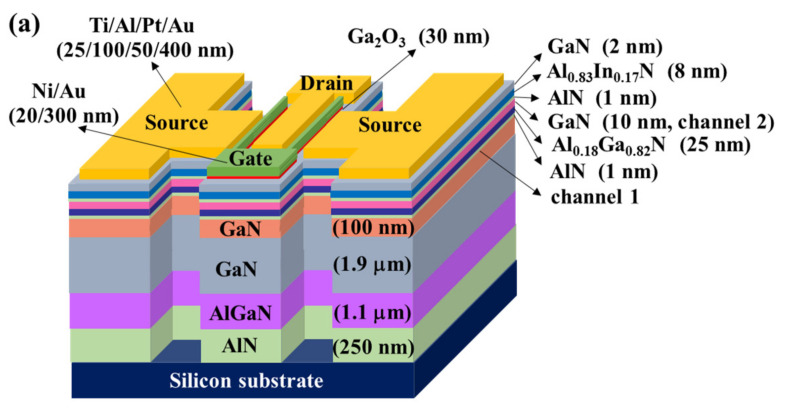
Epitaxial structure and three-dimensional schematic configuration of double-channel MOSHEMTs with (**a**) planar channel and (**b**) multiple-mesa-fin-channel array.

**Figure 2 materials-15-00042-f002:**
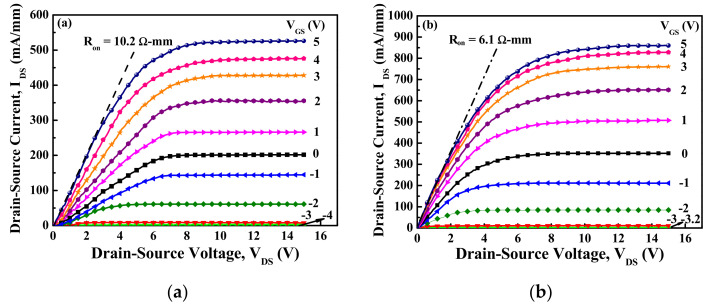
Typical drain–source current−drain–source voltage characteristics of double-channel MOSHEMTs with (**a**) planar channel and (**b**) multiple-mesa-fin-channel array [29].

**Figure 3 materials-15-00042-f003:**
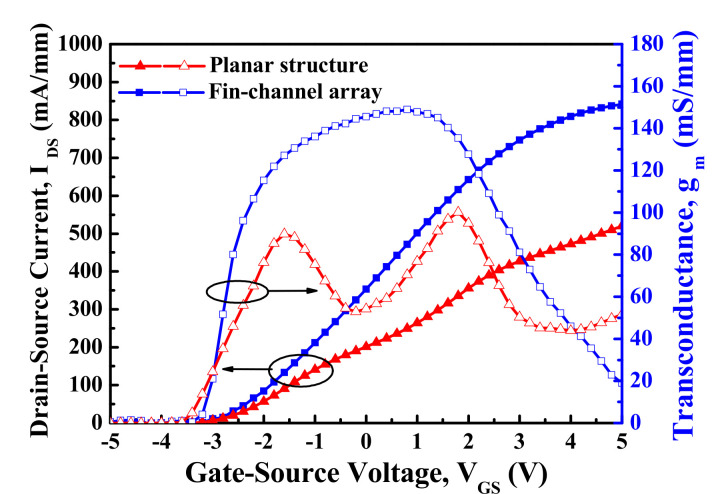
Drain–source current−gate–source voltage characteristics and extrinsic transconductance/gate–source voltage characteristics of double-channel MOSHEMTs operating at drain–source voltage of 10 V.

**Figure 4 materials-15-00042-f004:**
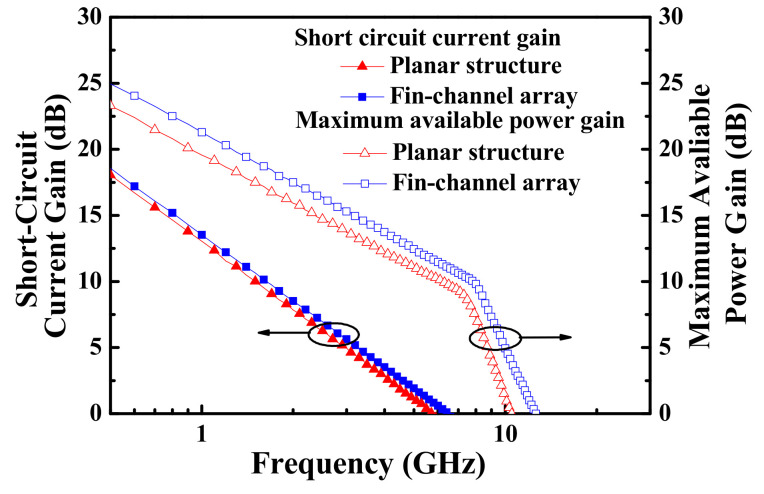
Short-circuit current gain and maximum available power gain as a function of frequency of double-channel MOSHEMTs.

**Figure 5 materials-15-00042-f005:**
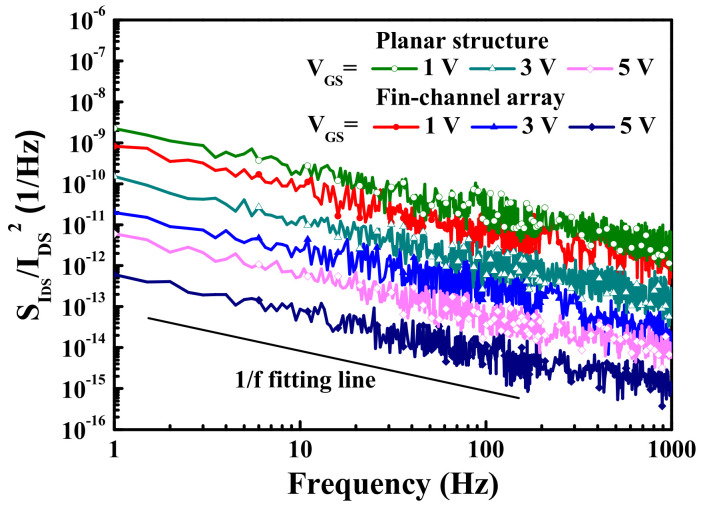
Frequency-dependent normalized noise power density spectra of double-channel MOSHEMTs operating at drain–source voltage of 1 V.

**Table 1 materials-15-00042-t001:** Performance comparisons of the double-channel MOSHEMTs with planar channel and multiple-mesa-fin-channel array.

MOSHEMTs	Planar Channel	Multiple-Mesa-Fin-Channel Array
Characteristics		
Drain–source current, I_DS_ at V_DS_ = 10 V and V_GS_ = 5 V	520.0 mA/mm	842.7 mA/mm
on-resistance, R_on_ at V_DS_ = 0 V and V_GS_ = 5 V	10.2 Ω-mm	6.1 Ω-mm
Transconductance, g_m_ at V_DS_ = 10 V	89.8 and 100.1 mS/mm	148.9 mS/mm
Threshold voltage, V_th_ at I_DS_ = 1 μA/mm	−3.8 V	−3.2 V
Unit gain cutoff frequency, f_T_	5.7 GHz	6.5 GHz
Maximum oscillation frequency, f_max_	10.5 GHz	12.6 GHz
Hooge’s coefficient, α	7.50 × 10^−^^5^	6.25 × 10^−^^6^

## Data Availability

The data presented in this study are available on request from the corresponding author.

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
