# Peer review of "Performance Comparison of Lattice-Matched AlInN/GaN/AlGaN/GaN Double-Channel Metal–Oxide–Semiconductor High-Electron Mobility Transistors with Planar Channel and Multiple-Mesa-Fin-Channel Array"

_materials, 2021, doi:10.3390/ma15010042_

Round 1

Reviewer 1 Report

The article propose not much more than it is presented in ref. [24] and references therein. Even experimental data for multiple-mesa-fin-channel array is taken from ref [24]. The main advantage of the present article is that the studies are done on the same heterostructure. But the differences in performance in-between the two constructions are very small, and its significance is not discussed. The main drawback of the article is the comparison of only two transistors with two different constructions. There is nothing in the article how statistically important are those differences.

English must be corrected as there are many errors. For example in lines:

33 (system -> systems)

Sentence in lines 40-41

Sentence in lines 54-55

98 (content -> contact)

123 (fi -> fin)

147 (pulsed -> pushed?)

155 (Using -> We used?)

172 (decreased -> decreases)

212 (challenge -> challenging?)

The conclusions are not only conclusions from the current article but also a discussion of the results and parts that should be included in the introduction as the current knowledge is discussed in the conclusion section.

Reviewer 2 Report

Multi-channels HEMT is promising device structure and gets highly intensive attentions. In this paper, authors combined the double-channel devices with multiple-mesa-fin-channel to greatly improve the transconductances, on-resistance and frequency properties. I recommend to accepting it after minor revisions. Some comments are listed as below:

1. Authors took the self-consistent calculation to calculate the energy band structure and charge distribution. Please draw the energy band diagram in the paper.

2. In fig.1, please indicate the channel 2. BTW, it is better to give a sketch map about the multiple-mesa-fin-channel arrays.

3. Hall measurements are performed to measure the 2DEG mobility and density. This value is an equivalent value in double channel 2DEG. It should be pointed out. BTW, it is possible to evaluate the contribution of different layers?

4. Please consider to cite some new literatures about HEMT, for example, Nature Electronics 4, 595 (2021), Nature Communications 11, 326 (2020).

5. Some units’ formats are missing during the document conversion. Please check them.  

Reviewer 3 Report

Comments:

The studies in the present work focus on the growth of the Al0.83In0.17N / GaN / Al0.18Ga0.82N / GaN epitaxial layers used to fabricate electron mobility transistors. Using a metal-organic chemical vapor deposition (MOCVD) system, dual-channel metal oxide semiconductors (MOSHEMTs) were cultured on silicon substrates using a metal-organic chemical vapor deposition (MOCVD) system.
The main criticism is concerning the way the manuscript is presented. It is more like a communication article than a standard article.
Below are my suggestions:
1) improve the introduction by inserting the importance and innovation of this work;
2) what is the importance and relevance of the device studied for the scientific community and industry? This point was not very clear!
3) Materials and methodos should be improved.
4) There was no structural characterization of the material to make the work more robust.
4) the most critical point to be improved is the discussion.
5) The conclusion must be improved.
6) The references used are very poor, lacking more references.
7) The way the manuscript was presented does not bring innovation to the area.
Therefore, my suggestion is to reject the article in its current form and, after the necessary modifications, the work should be re-evaluated.

Reviewer 4 Report

The paper is all about comparing two devices where they claim that "Multiple-Mesa-Fin-Channel Array" MOSHEMT has an edge over the planer channel in terms of performance.   However, I found some issues with the paper and the information provided.  

  1. Epitaxial Structure of Multiple-Mesa-Fin-Channel Array MOSHEMT isn’t provided.
  2. Images of the fabricated devices aren’t provided.
  3. Smoothness of the curves in Fig. 2 seems too good to be true for experimental devices.
  4. They claim that 2DEG is formed in simulation in two interfaces using 1D Schrodinger-Poisson solver but didn’t provide the output figure.
  5. Not being innovative enough, some conclusions are obvious. 

Round 2

Reviewer 3 Report

The authors answered the questions satisfactorily and made the recommended modifications. Thus the article is ready to be published.

Author Response

Thanks reviewer’s comments.

Reviewer 4 Report

 Accept in present form

Author Response

Thanks reviewer’s comments.
